# Astrocyte-Derived Exosomes Differentially Shape T Cells’ Immune Response in MS Patients

**DOI:** 10.3390/ijms24087470

**Published:** 2023-04-18

**Authors:** Piotr Szpakowski, Dominika Ksiazek-Winiarek, Joanna Czpakowska, Mateusz Kaluza, Marta Milewska-Jedrzejczak, Andrzej Glabinski

**Affiliations:** Department of Neurology and Stroke, Medical University of Lodz, Zeromskiego 113 Street, 90-549 Lodz, Poland

**Keywords:** astrocytes, exosomes, multiple sclerosis, T-cells

## Abstract

Astrocytes, the most abundant group of glia cells in the brain, provide support for neurons and indicate multiple various functions in the central nervous system (CNS). Growing data additionally describe their role in the regulation of immune system activity. They exert their function not only by direct contact with other cell types, but also through an indirect method, e.g., by secreting various molecules. One such structure is extracellular vesicles, which are important mediators of crosstalk between cells. In our study, we observed that the impact of exosomes derived from astrocytes with various functional phenotype differently affect the immune response of CD4+ T cells, both from healthy individuals and from patients with multiple sclerosis (MS). Astrocytes, by modulating exosome cargo, impacts the release of IFN-γ, IL-17A and CCL2 in our experimental conditions. Considering the proteins concentration in cell culture supernatants and the cellular percentage of Th phenotypes, it could be stated that human astrocytes, by the release of exosomes, are able to modify the activity of human T cells.

## 1. Introduction

Astrocytes are the most prevalent glial cells in the central nervous system (CNS) [1]. Previously, they were considered mainly as a trophic and mechanical support for neuronal cells. Lately, it has been widely accepted that they play multiple roles in the CNS, e.g., they support the blood–brain barrier (BBB), control the ion balance and modulate some neuronal functions [2]. They may also act as antigen-presenting cells (APC), regulating immune responses within the CNS [3]. Astrocytes triggered by various stimuli may participate in neuroinflammation development [4,5,6]. This process results in the production of cytokines, chemokines, reactive oxygen species and the release of some secondary messengers [7]. As secretory cells, astrocytes release large amounts of vesicles, such as extracellular vesicles (EVs). Their content differs depending on the character of the stimulants presented in the astrocytes’ microenvironment [8].

EVs are membrane-derived vesicles released by almost all cell types. They are involved in intercellular communication locally and systematically [9,10]. EVs were initially thought of as waste carriers. However, now their role in exchanging materials between cells, signal transduction, maintaining cellular homeostasis and the spread of pathological changes has been confirmed [11,12,13]. They participate in the regulation of various biological processes through the transport of their cargo from donor cells to recipient ones [14].

Recent studies suggest that astrocytes release EVs to communicate not only with other brain cells, but also with peripheral immune cells. Exosomes derived from CNS-resident cells were found in the circulation, which indicates their ability to cross the BBB [15]. In vitro experiments further support their potential for transmigration across this barrier. Such ability results in the propagation of a neuroinflammatory or neuroprotective response to the periphery [16,17].

The bi-directional communication between the CNS and the immune system is an important aspect of the pathogenesis of multiple sclerosis (MS). MS is an immune-mediated disorder in which inflammatory processes take place in the periphery, BBB and CNS parenchyma. This disease is one of the main causes of progressive disability in young adults. Thus, the detailed studies analyzing the pathomechanisms of MS are of great importance to increase our knowledge about its development and repair processes. The aim of our study was to assess the impact of EVs released by astrocytes with various functional phenotypes on the CD4+ T immune response.

## 2. Results

### 2.1. CD63-Positive-Exosomes Production by Astrocytes

Tetraspanin CD63 is one of the proteins characteristic of exosomes [18]. The presence of CD63 was confirmed in the lysates of exosomes samples (Figure 1). Additionally, the exact number of exosomes was determined with an acetecylcholinesterase activity-based assay (FluoroCet, System Bioscience, Palo Alto, CA, USA).

### 2.2. Th1 and Th17-Characteristic Cytokines Production by CD4+ T-Cells Stimulated with Astrocytic Exosomes

CD4+ lymphocytes from MS-affected individuals significantly decreased IFN-γ production in the presence of exosomes derived from resting or IL-10-stimulated human astrocytes (Figure 2A). Such an effect was not observed in cell cultures from non-MS patients (Figure 2B). IL-17A production was reduced in cell cultures stimulated with exosomes derived from resting astrocytes (Figure 2D), in comparison to production of this cytokine by T-cells alone or T-cells cultured with exosomes derived from IL-10- or TGF-β-primed astrocytes. Such an effect was observed only in a group of non-MS subjects. We did not observe any significant differences in IL-17A or IFN-γ production between T cells obtained from healthy donors and MS patients (Appendix A).

### 2.3. CCL2 Production by Astrocytic Exosome-Stimulated CD4+ T-Cells

CCL2 production by CD4+ T-cells was upregulated in cell cultures pulsed with exosomes derived from astrocytes primed with proinflammatory cytokines, and compared with spontaneous CCL2 release. Such an effect was observed for cells from MS-affected persons and healthy donors (Figure 3). Furthermore, the CCL2 concentration in the culture media from CD4+ T-cells obtained from healthy donors was significantly lower in the presence of exosomes produced by IL-10- or TGF-β-primed astrocytes by comparison with its production by T-cells primed by exosomes from proinflammatory astrocytes (Figure 2B).

### 2.4. Flow Cytometry Measurements

Flow cytometry analysis of the composition of T-cells producing IFN-γ (Th1), IL-17A (Th17), IL-4 (Th2), and IL-10 (Treg) revealed no visible changes in the percentage of Th1 and Treg cells in response to various exosomes for both MS patients and control subjects without autoimmune disease. What is interesting is that we observed significant difference in the percentage of IL-17-positive T-cells in the MS patients group between cells pulsed with exosomes from resting astrocytes and exosomes isolated from TGF-β primed astrocytes (Figure 4A). Cells from donors without autoimmune disease, in response to exosomes produced by resting astrocytes, increased the number of IL-4-producing cells as compared to cells cultured without exosomes. On the contrary, exosomes isolated from TGF-β-primed astrocytes decreased the percentage of IL-4-producing cells, when compared to non-stimulated cells or cells stimulated with exosomes from resting astrocytes. Differences are at the border of statistical significance (*p* = 0.07). Flow cytometry scatter plots and a description of the gating strategy are presented in the supplementary material (Appendix A). We have also analyzed the percentage of cells producing both IL-17A and IFN-γ. We did not observe significant difference in % of CD4^+^ IL-17A^+^ IFN- γ^+^ cells between lymphocytes pulsed with different exosomes. However, difference between cells from MS-patients and control subjects were observed (Appendix A). 

## 3. Discussion

In recent years, the role of exosomes has been highlighted in the context of cell-to-cell communication. These vesicles belong to the heterogeneous group of extracellular vesicles (EVs). EVs are classified into three main types based on their size and morphology—macrovesicles or microparticles, and apoptotic bodies and exosomes. The first group is obtained from plasma membrane and are 100 to 1000 nm in diameter. The apoptotic bodies are obtained from dying, apoptotic cells and range from 1000 to 5000 nm [19]. Exosomes are 50 to 150 nm in diameter, and are stored within multivesicular bodies (MVBs). They are released into the extracellular space through the fusion of MVBs with the plasma membrane [20]. These small bilayer-enclosed vesicles are released from almost all cell types [9,10]. They regulate various biological processes via the transfer of their cargos to the recipient cells [14]. Their cargo encompasses nucleic acids, proteins, lipids, amino acids, metabolites, glycoconjugates, cytosolic and cell-membrane proteins [14,21]. It has been proven that exosome cargo is remarkable stable and may reflect the physiological/pathological state of the parent cells. Moreover, it was shown that different cell types can secrete exosomes with different markers, which enables the identification of their cell source [22].

Astrocytes are the most numerous glial cells playing a pivotal role in maintaining the architecture and function of the CNS [1,23,24]. Originally, they were considered as supporting cells for neurons. Growing evidence has pointed to the fact that their role is far more complex, both in health and disease states. Astrocytes express a wide range of receptors: they recycle neurotransmitters, form a tripartite synapse [25], participate in the formation and maintenance of the BBB, and act as a structural support to neurons, which makes them important regulatory cells in the CNS [1,26,27,28,29,30,31,32]. Astrocyte reactivity is regulated by various factors from diverse sources, such as age-mediated intracellular alterations within molecular pathways, molecules such as cytokines, growth factors, pathogens, or environmental toxins [2]. These cells have the ability to change their functional phenotype from pro-inflammatory to anti-inflammatory and vice versa according to the stimuli presented in their microenvironment [4,5,6]. Reactive astrocytes contribute to pathological changes observed in various CNS disorders [33]. However, there is also evidence that they may exert a protective role in some neurological diseases [34,35].

One of the modes of action by which astrocytes affect the function and phenotype of other cell types is via the secretion of EVs [36,37,38,39]. Their effect depends on the functional state of astrocytes. At a normal/resting state, their EVs have been shown to regulate neurogenesis, angiogenesis and neuroprotection in vitro [33,40,41,42,43]. It has been shown that naive astrocytes shed EVs with a neuroprotective cargo, e.g., the fibroblast growth factor-2, the vascular endothelial growth factor and apolipoprotein-D [40,41]. Under stimulation in pathological or aging conditions, activated astrocytes secrete EVs with an altered cargo content, e.g., proteins and miRNA [8,33,36]. Stimulation with pro-inflammatory factors, such as IL-1β or TNFα results in the secretion of EVs that participate in inflammatory signal transmission, neurite extension and branching inhibition, and a reduction in neuronal excitability [8,33,36]. It was also shown that they participate in the spreading of aberrant proteins [44,45]. The presence of anti-inflammatory cytokines in the microenvironment, such as IL-10, results in astrocyte-derived EVs that contain proteins, such as heat shock proteins, synapsin 1, selected microRNAs, and glutamate transporters [34,35,46], promoting neuronal plasticity, neurite outgrowth and neuronal survival [8]. This suggests that astrocyte-derived EVs play not only a detrimental, but also a protective role in CNS disorders, depending on the character of the stimulating factors.

Astrocytes, as part of the BBB, are one of the first CNS-resident cells that interact with blood-derived leukocytes entering the brain during inflammatory processes [47]. It has been shown that reciprocal interaction between astrocytes and leukocytes regulates the transmigration of the latter into the CNS [48]. Thus, it has been concluded that astrocyte-derived EVs regulate not only neighboring CNS cells, but also participate in communication between the CNS and peripheral immunity. Under pro-inflammatory conditions, induced by IL-1β, it was shown that proteins and miRNAs from astrocyte EVs promote peripheral acute cytokine response (ACR) [37]. This can be achieved because EVs transmigrate across the BBB [15]. EVs elevate the immune response by increasing nuclear factor kappa B activity and elevating inflammatory cytokine expression in the liver. In vivo studies have shown that such EVs also promote leukocyte transmigration into the CNS, further augmenting neuroinflammation [37]. Despite the above-mentioned studies, little is still known about the effect of exosomes released by astrocytes on peripheral immune cells. One of the diseases in which the relationship between CNS-resident cells and the peripheral immune system plays a pivotal role is multiple sclerosis (MS).

MS is one of the leading CNS disorders affecting young adults [49]. It is characterized by multifocal inflammation, demyelination, and neuronal loss, which results in progressive disability in affected individuals [50]. Despite numerous studies, the etiopathogenesis of this disorder is still unknown. It has been stated that genetic, epigenetic and environmental factors are together responsible for disease development [50,51]. As was mentioned earlier, the important aspect of MS is the communication and relationship between CNS-resident cells and peripheral immunity.

With the growing evidence in the field of EVs biology and function, it is stated that EVs are important mediators of both pathological and reparative processes in MS [52,53], including communication between the CNS and the immune system. It has been observed that exosomes can activate or suppress the immune system by regulating the function of various immune cells [54,55]. Exosomes containing components originating from parent cells interact with recipient cells, changing their function and phenotype features [56].

It is stated that two subpopulations of Th cells have a detrimental role in MS development—namely Th1 and Th17 cells. Th17 lymphocytes are said to be implicated in the initiation of pathogenic immune response in CNS. IL-17 family cytokines were shown to increase the permeability of BBB and induce chemokine production from the brain endothelium to exacerbate immune cells transmigration into the site of inflammation [57]. Th1 cells are important players in maintaining inflammatory reactions in the CNS. They are the main source of IFN-γ, which activates the local immune responses of tissue-resident cells. In our study, we did not observe any changes in the percentage of IFN-γ-producing T cells (Th1) in any of the studied groups. However, the addition of astrocyte-derived exosomes altered the production of this cytokine in MS patients. Exosomes from unstimulated/resting and IL-10-primed astrocytes significantly reduced the IFN-γ level. This suggests the potential new pathway of the astrocyte impact on the immune response shape.

Contradictory results were obtained for IL-17A, where T-cells culturing in the presence of exosomes changed the level of this cytokine only in non-MS patients. Exosomes from resting/unstimulated astrocytes lowered the expression level of IL-17A. Surprisingly, we noticed elevated levels of this cytokine after co-culture with exosomes from IL-10- and TGF-β-stimulated astrocytes. This may suggest the unbalanced character of astrocytes stimulated with potentially anti-inflammatory cytokines. In our previous work, we reported the production of IL-1β by astrocytes in response to IL-10 [8].

Overall, this pointed to the assumption that various functional phenotypes of astrocytes result in the secretion of exosomes with an altered cargo that may have a variable impact on the secretory activity of T-cells.

CCL2, previously known as a monocyte chemoattractant protein-1 (MCP-1), is a chemokine expressed by a wide variety of cells, such as endothelial cells, smooth muscle cells, fibroblasts, epithelial cells, astrocytes, T-cells, as well as by myeloid cells [58,59,60,61]. Its expression is often induced by inflammatory stimuli, such as IL-1, IL-4, IL-6, TNF-α, TGF-β, IFN-γ and others [62,63,64,65,66]. Although CCL2 is a chemoattractant mainly for monocytes, it was shown that it can also attract other cell types, e.g., T-cells, B-cells, dendritic cells, macrophages etc., according to specific conditions [67,68,69,70,71,72,73]. CCL2 signals mainly through the binding to and activation of CCR2 [74]. It has been shown that the role of CCL2 is not only limited to the attraction of immune cells. This chemokine may also control monocyte adhesion by modulating integrin expression and localization. It may also play a role in monocyte activation to produce inflammatory cytokines [75,76]. Moreover, it was observed that CCL2 enhances the survival of macrophages and neutrophils [77,78]. CCL2 has been widely studied in the context of MS and its animal model—experimental autoimmune encephalomyelitis (EAE). The high expression level of this chemokine was observed in MS lesions. According to results obtained in EAE models, it was concluded that CCL2 released by glial cells significantly impacts the transmigration of inflammatory cells (monocytes, T cells and dendritic cells) into the CNS [79,80]. In our work, we observed that human astrocytes through secreted exosomes may regulate CCL2 release in T-cell cultures. Inflammation in the CNS linked with microglia activation and exposition of astrocytes to proinflammatory factors (TNFα, IL-1a, C1q) may lead to the secretion of exosomes. This will, in turn, increase CCL2 production by infiltrating T-cells, resulting in exacerbation of the immune response by promotion of monocytes income, and additional T-cells recruitment. A similar effect was observed for cells from donors not affected with autoimmune disease, where exosomes from proinflammatory astrocytes increased CCL2 production, and the amount of CCL2 was significantly higher in T-cells culture exposed to exosomes from proinflammatory astrocytes, as compared to exosomes from TGF-β-primed astrocytes. However, the impact of exosomes on the activity of T-cells from MS patients seems to be not exactly the same as that on T-cells from healthy subjects. In fact, we did not observe any difference in CCL2 production between T-cells stimulated with exosomes from proinflammatory cytokines-primed astrocytes and exosomes produced by IL-10- or TGF-β-pulsed ones in the studied group, contrary to what was observed for non-MS patients.

In conclusion, our data suggest the role of astrocyte-derived exosomes in the regulation of T-cells secretory activity. This may suggest a new function for astrocyte cells, which seems to be implicated not only in the recruitment of immune cells by the regulation of blood–brain barrier permeability and chemokine release, but also in the modification of T-cells secretory activity through their exosome cargo. Because CNS–peripheral immune system communication is important in various neurological autoimmune diseases such as MS, and affects patients’ recovery, it is enormously important to deepen our knowledge of mechanisms regulating this process. Thus, the more detailed studies of the role of exosomes in shaping the interplay between immune and nervous system seem to be of great significance.

## 4. Materials and Methods

### 4.1. Astrocyte Cell Cultures

Human primary astrocytes were bought from ScienCell Research Laboratories (San Diego, CA, USA). Astrocytes were isolated from the cerebral cortex of a 22-week-old human donor. Cells were grown on poly-L-lysine-coated 75 cm^2^ culture flasks, in astrocyte medium supplemented with antibiotics (penicillin, streptomycin), astrocyte growth supplement and 2% FBS (all components from ScienCell Research Laboratories) in 5% CO_2_ and an increased-humidity atmosphere, at 37 °C. The culture medium was changed every three days, or daily for a culture confluency over 75%. When the culture reached 90% confluency, the cells were washed with Dulbecco’s PBS (DPBS, Corning, Corning, NY, USA), detached with 0.025% tripsin/EDTA (ethylenediaminetetraacetic acid) in DPBS and a 5%FBS/DPBS solution and centrifuged (150× *g*, 5 min, 20 °C).

### 4.2. Isolation of Exosomes from Astrocyte Cultures

DPBS-washed astrocytes were seeded into new culture flasks (107 cells/flask) and cultured in a medium containing FBS without exosomes (SBI System Biosciences, Palo Alto, CA, USA). Some 12 h later, the cells were stimulated with a cocktail of proinflammatory cytokines: IL-1a (10 ng/mL, R&D Systems, Ixonia, WI, USA), TNF-α (30 ng/mL, R&D Systems), C1q (400 ng/mL, R&D Systems). Alternative phenotypes of astrocytes were induced by IL-10 (10 ng/mL) or TGF-β (10 ng/mL) stimulation (both cytokines from R&D Systems). Astrocytes cultured without cytokines were also included. In our previous work, we characterized the secretory activity of astrocytes in response to these stimuli [81]. After a six-day culture, the media were collected and centrifuged (300× *g*, 10 min, 20 °C) and the supernatants were centrifuged again (16,000× *g*, 30 min, 20 °C). Next, the exosomes were precipitated (12 h, 4 °C) with exosome precipitation reagent (Exo-spin Buffer, CellGS, Cambridge, UK) and centrifuged (16,000× *g*, 1.5 h, 20 °C). The supernatants were removed and the pellets were resuspended in PBS and purified on exosome size-exclusion columns (Exo-spin, Cell Guidence Systems, Cambridge, UK) according to a protocol provided by the assay manufacturer. After purification, the exosomes were aliquoted and frozen for further procedures.

### 4.3. Exosomes Detection and Quantification

Parts of the exosome samples were lysed with RIPA lysis buffer (Merck, Rahway, NJ, USA) and protein concentrations in exosome lysates were determined with a BCA protein assay (Pierce, Appleton, WI, USA). Exosome quantification in samples was carried out with a Fluorocet kit (SBI, System Biosciences). The presence of exosomes in samples was also confirmed by Western blot assay.

#### Exosomes Detection by Western Blot

Samples denaturized (10 min, water bath 70 °C), with Bolt LDS sample buffer (Thermo Fisher Scientific, Waltham, MA, USA) were loaded into the gel rows (total protein load 2 μg in 40 μL/row). Electrophoresis was carried out on Bolt 4–12% Bis-Tris Plus gel in Bolt MES SDS running buffer (22 min, 200 V), under non-reducing conditions; protein standards were also included (Magic Mark XP, PAGE Ruler Plus Thermo Fisher Scientific). Proteins were transferred on to the PVDF membrane in Bolt Transfer Buffer containing 20% of methanol (1 h, 25 V). After the protein transfer, the membrane was washed, blocked and labelled with antibodies: anti-CD63 IgG1 mouse primary antibody (Cell GS; 0.5 µg/mL, overnight, 4 °C); HRP-conjugated secondary antibody donkey anti-mouse IgG (Thermo Fisher Scientific, 8 ng/mL, 1 h, 20 °C). Target proteins were detected with ECL detection system (Thermo Fisher Scientific, Dura West Atto), and visualized with a G-BOX imaging system (Syngene, Bengaluru, India).

### 4.4. Material from Human Donors

Blood donors were recruited from among patients in the Department of Neurology and Stroke, Central Veterans Hospital, Medical University of Lodz, diagnosed with a relapsing-remitting form of multiple sclerosis, without immunomodulatory treatment before the blood donation (10 individuals). The control group encompassed six patients suffering from headaches, without any autoimmune background, matched by age and gender within the study group. The study was conducted according to the principles of the Declaration of Helsinki and was approved by the Local Ethics Committee of the Medical University of Lodz. All study participants gave written consent to participation in the study. Blood samples were collected in a volume of 45 mL on an anticoagulant (S-Monovette, Sarstedt, Numbrecht, Germany, containing K_3_EDTAK_2_EDTA). In the study group, clinical symptoms of a relapsing-remitting form of MS were confirmed with MRI resonance and neurological diagnostics.

### 4.5. Human CD4+ T-Cells Separation

Blood samples drawn on anticoagulant-containing tubes (Sarstedt, Nümbrecht, Germany) were initially twice diluted with PBS (without Ca^2+^ and Mg^2+^). Peripheral blood mononuclear cells (PBMC) were separated from whole blood on a density gradient (Ficoll-paque plus); next, the CD4-positive T-cells were isolated using magnetic assay cell sorting (MACS positive selection, Miltenyi Biotec, Teterow, Germany). The cell number was determined by cell counting in a Bürker chamber.

### 4.6. T-Cell Cultures

After separation, human CD4+ T-cells were cultured on 48-well culture plates (106 cells/well) in a culture medium RPMI1640 (Corning), containing 10% FBS (exosomes depleted) and antibiotics (penicillin/streptomycin; Corning). Cultures were supplemented with exosomes obtained from human primary astrocytes cultured in resting, pro- or anti-inflammatory conditions. Exosome suspensions were added in a volume containing 108 exosomes. As a control, CD4+ T-cells were cultured in a medium alone. After 96 h, culture media from T-helper cells exposed to exosomes were collected, centrifuged (4000× *g*, 10 min, 20 °C), aliquoted and frozen for further measurements. A new cell culture medium containing brefeldin A (Golgi plug, BD Biosciences, Franklin Lakes, NJ, USA) was added, and T-cells were cultured additionally overnight.

### 4.7. Secretory Activity of T-Cells Exposed to Astrocytes-Derived Exosomes

The concentration of IFN-γ, IL-17A, IL-4, IL-10, and CCL2 in medium supernatants, collected from cultures of CD4+ T-lymphocytes exposed to different exosomes, was measured with ELISA (all ELISA tests from R&D Systems).

### 4.8. Analysis of the Impact of Exosomes from Different Astrocyte Phenotype on T-Cells Activity by Flow Cytometry

T-cells were collected from culture plates. In order to analyze the number of lymphocytes producing IFN-γ (Th1), IL-17 (Th17), IL-4 (Th2) and IL-10 (Treg), cell suspensions (50 μL) were stained extracellularly with antibody cocktails. All antibodies used for flow cytometry cells labelling are listed in Table 1. After extracellular staining, cells were washed twice (with ice-cold PBS/1% FBS), fixed with 4% cold formalin (20 min, 4 °C), washed, frozen in FBS/10% DMSO and stored for the flow cytometry measurements. Before the analysis, cells were thawed, washed twice, permeabilized for 15 min with ice-cold perm/wash buffer containing saponin (BD Biosciences). Next, the cells were stained intracellularly (30 min, 4 °C, in dark).

### 4.9. Statistical Analysis

Statistical analysis was performed with Statistica 13 Software. Comparisons between variables were performed with a non-parametric Wilcoxon signed-rank test.

## Figures and Tables

**Figure 1 ijms-24-07470-f001:**
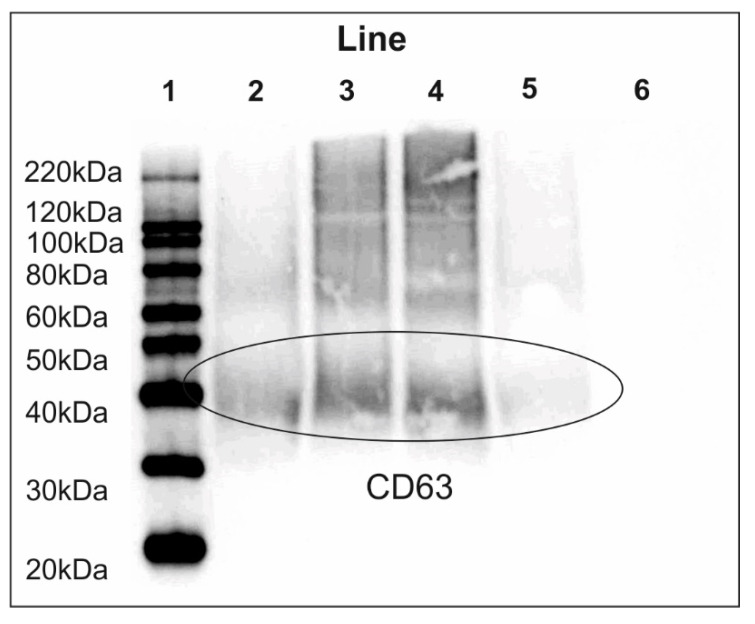
Western blot for human CD63 detection in astrocyte-derived exosome lysates. Exosomes were separated from supernatants of cell culture medium. Line 1—molecular marker (Invitrogen Magic Mark XP, Invitrogen, Waltham, MA, USA), lines 2–5: RIPA lysates of exosomes (2.0 µg of total protein/line) isolated from culture medium after human primary astrocytes stimulation with TGF-β (2), IL-10 (3), proinflammatory cytokines TNFα/IL-1a/C1q (4), culture medium alone (5). Line 6—cell lysate from astrocytes.

**Figure 2 ijms-24-07470-f002:**
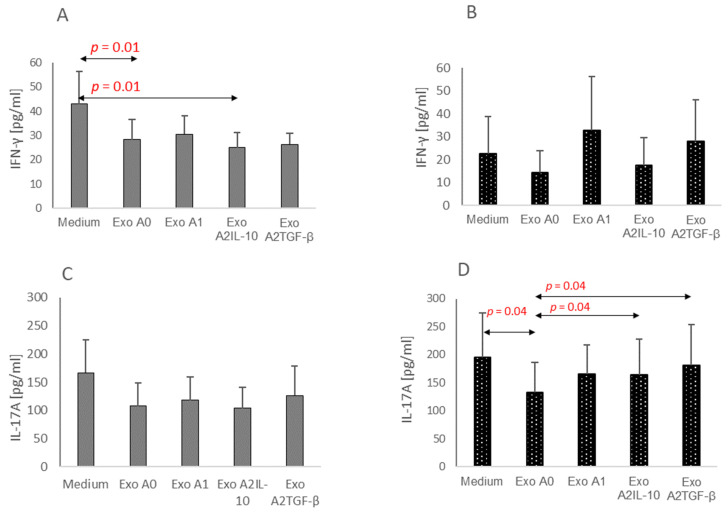
IFN-γ (**A**,**B**) and IL-17A (**C**,**D**) production in CD4+ T-cell cultures from multiple sclerosis-affected patients (**A**,**C**) and healthy donors (**B**,**D**) (with excluded autoimmune disease). After separation from PBMC, CD4+ T-cells from every donor were isolated by magnetic sorting, and cultured on 48-well culture plates (10^6^ cells/well) with exosomes isolated from non-stimulated (Exo A0), pro-inflammatory cytokines stimulated (Exo A1) and anti-inflammatory cytokines (IL-10, TGF-β) stimulated (respectively Exo A2IL-10, Exo A2TGF-β) human astrocytes. Data shown as mean ± SEM. Comparisons between variables were made with a Wilcoxon signed-rank test.

**Figure 3 ijms-24-07470-f003:**
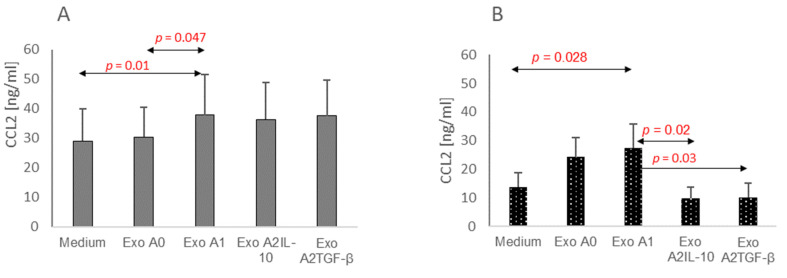
CCL2 chemokine production by CD4+ T-cells isolated from MS-affected patients (**A**) and healthy donors (**B**) (with excluded autoimmune disease). After separation from PBMC, every donor’s CD4+ T-cells were isolated by magnetic sorting, and cultured on 48-well culture plates (106 cells/well), with exosomes isolated from non-stimulated (Exo A0), pro-inflammatory cytokines stimulated (Exo A1) and anti-inflammatory cytokines’ (IL-10, TGF-β) stimulated (respectively Exo A2IL-10, Exo A2TGF-β) human astrocytes. Data shown as mean ± SEM. Comparisons between variables were made with Wilcoxon signed-rank test.

**Figure 4 ijms-24-07470-f004:**
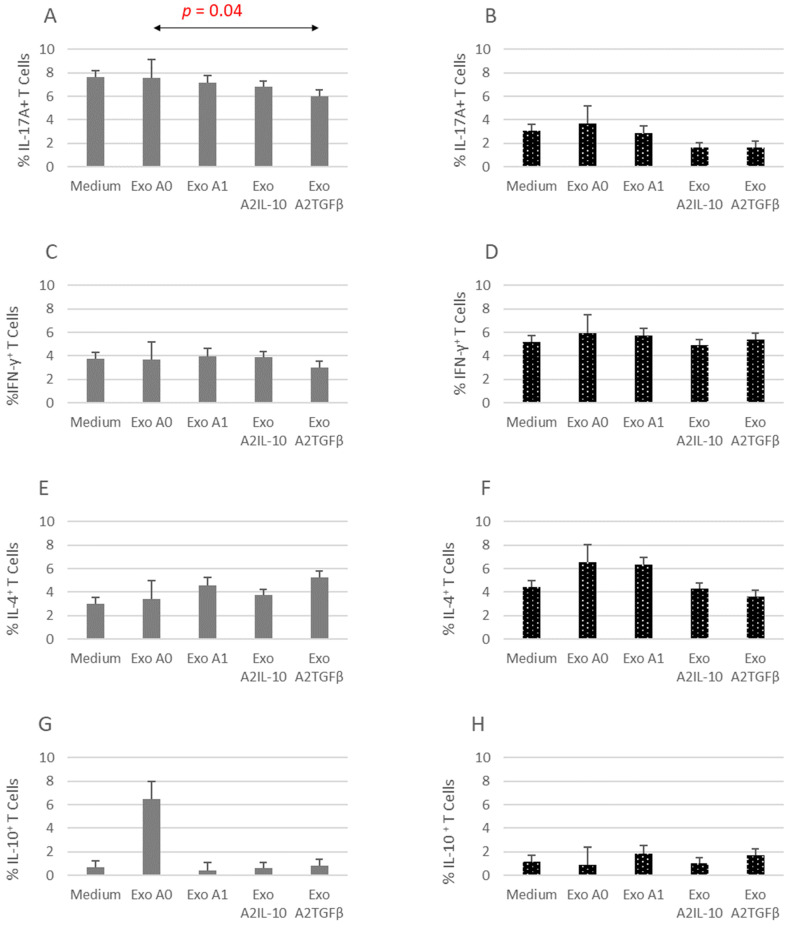
Results of flow cytometry analysis of cell composition. Mean percentages of CD3^+^ CD4^+^ cells producing IL-17 (**A**,**B**), IFN-γ (**C**,**D**), IL-4 (**E**,**F**) and IL-10 (**G**,**H**) cytokines are presented. Data for MS patients are in graphs with filled bars (7 persons), results for non-MS cells are in white dotted bars (4 persons). After 96 h of T-cell cultures in presence of astrocyte-derived exosomes or medium alone, culture media were collected, cells were suspended in new medium containing brefeldin A for 12 h; next, cells were harvested and stained for flow cytometry. Cells were analyzed as a % of CD3^+^ CD4^+^ cells positive for the individual cytokines. Data were shown as mean ± SE. Comparisons between variables were made with non-parametric Wilcoxon signed-rank test.

**Table 1 ijms-24-07470-t001:** Antibodies used for flow cytometry staining.

	Antibody Set 1	Antibody Set 2
Extracellular staining:	Anti-CD3 Pacific Blue	Anti-CD3 Pacific Blue
	Anti-CD4 FITC	Anti-CD4 FITC
		Anti-CD25 PerCP
Conditions:	30 min, 4 °C, in dark,	
Intracellular staining:	Anti-IFN-γ BV510	Anti-FoxP3 PE
	Anti-IL-17A PE	Anti-IL-10 APC
	Anti-IL-4 APC	
Conditions:	Permeabilization: 15 min, 4 °C, in dark	
	Staining: 30 min, 4 °C, in dark	

## Data Availability

The data presented in this study are available on request from the corresponding author. The data are not publicly available due to privacy.

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
