# Peer review of "Astrocyte-Derived Exosomes Differentially Shape T Cells’ Immune Response in MS Patients"

_ijms, 2023, doi:10.3390/ijms24087470_

Round 1
Reviewer 1 Report
In the current study, Szpakowski et al have investigated how extracellular vesicles released by astrocytes affect CD4+ T cell phenotype in MS and non-MS patients. The manuscript is well written, precise and the language is in good shape. Even though the introduction is short and doesn’t give enough background however this is corrected in discussion section where authors have provided relevant background introduction and references to the current study. I have only few comments that I believe would make this manuscript even better.
· Did the authors try culturing astrocytes in media containing IL-6 and TGF-b or IL-6 alone without TGF-b ? There are reports that astrocytes produce abundant amount of IL-6 during neuroinflammation. It will be interesting to know if astrocyte derived IL-6 either promotes Th17 phenotype or leads to STAT1 activation instead?
· During the flow analysis of cytokine production from CD4+ T cells cultured with exosomes derived from astrocytes cultured in various stimulations, did authors look at double producers of CD4+IL17+IFNg+ in each condition? Was there any GMCSF production from CD4+ T cells ?
Author Response
In the current study, Szpakowski et al have investigated how extracellular vesicles released by astrocytes affect CD4+ T cell phenotype in MS and non-MS patients. The manuscript is well written, precise and the language is in good shape. Even though the introduction is short and doesn’t give enough background however this is corrected in discussion section where authors have provided relevant background introduction and references to the current study. I have only few comments that I believe would make this manuscript even better.
Reviewer
Did the authors try culturing astrocytes in media containing IL-6 and TGF-b or IL-6 alone without TGF-b ? There are reports that astrocytes produce abundant amount of IL-6 during neuroinflammation. It will be interesting to know if astrocyte derived IL-6 either promotes Th17 phenotype or leads to STAT1 activation instead?
Answer:
We are thankful for this suggestion. In fact, investigation of the impact of IL-6 or IL-6 with TGF-β on the biology of astrocytes and their ability to shape Th17 seems to be verry interesting. In this work our choice of the stimulants was supported by the literature, describing the TNF-α, IL-1a and C1q cytokines as a major inducers of pro-inflammatory phenotype of astrocytes. It is quite possible that in our conditions astrocytes were producing IL-6 in response to proinflammatory stimuli, as was already shown that astrocytes are the major source of this cytokine in CNS (https://doi.org/10.1523/JNEUROSCI.19-13-05236.1999). However, our goal was to investigate and compare the impact of EVs secreted by astrocytes, rather than the impact of other molecules present in cell culture medium after various stimulatory conditions. Nevertheless, we will consider such stimulation with IL-6 or IL-6/TGF-β in our future experiments.
Reviewer1
During the flow analysis of cytokine production from CD4+ T cells cultured with exosomes derived from astrocytes cultured in various stimulations, did authors look at double producers of CD4+IL17+IFNg+ in each condition? Was there any GMCSF production from CD4+ T cells?
Answer:
We have performed the flow cytometry analysis of the double producers of CD4+IL17+IFN-γ+ in our experimental conditions, and we have observed significant differences between T cells from healthy donors and MS-affected patients. However, exosomes were not able to significantly change the percentage of double positive T cells in each condition (gating strategy and figure included in the supplementary material Supplement 3).
We did not measure the GMCSF production in cell cultures.
Reviewer 2 Report
In this manuscript the authors investigated the impact of exosomes derived from astrocytes with various functional phenotype, differently affect the immune response of CD4+ T cells, both from healthy individuals and patients with MS. Some concerns and suggestions are listed as below:
The English of this manuscript should be edited. For example, there are two 'astrocytes' in line 13. In addition, lines 60-63 can be removed from the manuscript.
Electron microscopy should be used for measuring exosomes.
In Figure 2, differences between A and B (C and D) should be marked in the figure.
In Figure 2, the variation was big. How did you interpret the results P=0.04 (Figure 2D)? In line 83, Fig. D should be Fig. 2D.
How did you confirm that these exosomes come from astrocytes, rather than other glial cells?
I wonder if astrocytes (rather than astrocyte-derived exosomes) have any effects on T cells from MS patients and controls.
Apart from cytokine production, other functions of T cells should also be performed.
FACS scatter plots should be provided (Figure 4).
Rescue experiments were lacking in this study. For example, exosome inhibitors.
I wonder if any DMTs may have effects on astrocyte-derived exosomes.
How astrocyte-derived exosomes come into the circulation? I wonder if they have any effects on immune cells (T cells) in the CNS.
In Figure 4, no need to mention p=0.07.
Components of astrocyte-derived exosomes should be investigated in details. What about their targets in T cells?
Author Response
Comments and Suggestions for Authors
In this manuscript the authors investigated the impact of exosomes derived from astrocytes with various functional phenotype, differently affect the immune response of CD4+ T cells, both from healthy individuals and patients with MS. Some concerns and suggestions are listed as below:
Reviewer2:
- The English of this manuscript should be edited. For example, there are two 'astrocytes' in line 13. In addition, lines 60-63 can be removed from the manuscript.
Answer:
The line 13 has been corrected. Lines 60-63 has been removed. The manuscript will be edited by the native speaker.
Reviewer2:
Electron microscopy should be used for measuring exosomes.
Answer:
We agree with the reviewer that electron microscopy will add additional confirmation of the character of EVs isolated in this study. However, we have used method of isolation based on concentration of sample and size exclusion chromatography (SEC), which provide isolation of particles ranging in size from 30 nm to 250 nm. This method of isolation is mentioned in the work by Thery and Witwer, 2018 (doi.org/10.1080/20013078.2018.1535750) as one of the highly specific. Moreover, we have analysed the presence of one of the markers of exosomes - tetraspanin CD63, present in exosomes (doi:10.1073/pnas.1521230113).
Reviewer2:
In Figure 2, differences between A and B (C and D) should be marked in the figure.
Answer:
Comparison between groups with U Mann-Whitney test did not indicate any significant difference in cytokines release between T-cells from healthy persons and MS-affected ones. Figures are included in the supplementary materials (Supplement 1).
Reviewer2:
In Figure 2, the variation was big. How did you interpret the results P=0.04 (Figure 2D)? In line 83, Fig. D should be Fig. 2D.
Answer:
The variation was big, what was caused by the individual variability. Cells from different donors have different baseline secretory activity, what may be impacted by numerous factors, like diet, physiological condition, etc.
According to Fig.2D we observed significant reduction in IL-17 release from T cells’ cultures in response to exosomes from resting astrocytes, what may be linked to the maintaining of strong anti-inflammatory environment in the brain. It is possible that astrocytes via their extracellular vesicles’ release contribute to such environment. This effect is observed only in T-cells from healthy donors (probably less activated as compared to T-cells from MS patients).
We have corrected the figure reference.
Reviewer2:
How did you confirm that these exosomes come from astrocytes, rather than other glial cells?
Answer:
In our experiments we have used commercially available human primary astrocytes. The characteristics of astrocytes were provided with cells. Provider has tested cells for astrocytes’ specific markers (GFAP). Moreover, we have confirmed the characteristic of these cells by analysis of GFAP by flow cytometry in our previous work (10.3390/biomedicines10081769).
Reviewer2:
I wonder if astrocytes (rather than astrocyte-derived exosomes) have any effects on T cells from MS patients and controls.
Answer:
We agree with the reviewer that such experiments will add valuable information about T cells and astrocytes interactions. However, there are technical issues making it difficult. The major obstacle is the immune compatibility between astrocyte cells and lymphocytes. Cells should come from the same donor. Use of proper culture medium for both cell types is also problematic in order to avoid non-specific activation of cells.
Moreover, the main aim of our study was to analyze the potential of astrocyte-derived exosomes, as information carriers having the ability to alter the functional phenotype of T cells. In our experimental conditions observed effect was only due to exosomes, as we do not co-culture T cells with astrocytes.
There are several works describing the astrocytes impact on T cells’ function. Astrocytes role is examined mainly in the context of the regulation of blood-brain barrier (BBB) transmigration of T cells. This process is modulated via direct micro-anatomical interactions between these two cell types and by various chemokines release (doi: 10.3389/fncel.2013.00058; doi.org/10.1172/JCI91301). Moreover, the direct contact between astrocytes and T cells depends on formation of immunological synapse (IS), where the signal is transduced via interactions between surface co-stimulatory molecules and proteins secreted to the synaptic space (DOI: 10.1084/jem.20060420; DOI: 10.1016/j.smim.2005.09.002; doi.org/10.3390/app11188557).
Reviewer2:
Apart from cytokine production, other functions of T cells should also be performed.
Thank You for your suggestion. Our goal was to investigate the impact of astrocytes via secreted exosomes on the CD4+ T-helper cells as cell type directly implicated in the induction and maintaining of inflammation in CNS. The secretory activity of Th1 and Th17 cells is critical for inflammatory process. What is more, other functions of T cells are the results of their secretory activity. Additional investigation of cytotoxic properties of CD4- and CD8-positive cells would be interesting and important, however we were limited by the number of T-cells necessary for our stimulation procedure. Results presented in this work have suggested that exosomes derived from astrocytes have the ability to change some aspects of T cells secretory activity.
Reviewer2:
FACS scatter plots should be provided (Figure 4).
Answer:
We have used a flow cytometry to identify functional T-cells subpopulations. We have added the flow cytometry scatter plots and description of gating strategy in supplementary materials (Supplement 2.)
Reviewer2:
Rescue experiments were lacking in this study. For example, exosome inhibitors.
Answer:
We agree that usage of exosome inhibitors will add important information to the field of astrocyte’s release of exosomes, to confirm that observed effect is not related to other vesicles. However, in our study design we have utilized the method of isolation based on concentration of sample and size exclusion chromatography (SEC), which is highly specific. We have confirmed the exosomes’ presence by two separate methods. What is more, we have used exosomes alone in T cells cultures, not astrocytes’ cell culture supernatants or co-culturing. That is the reason why we used culture medium alone as a negative control.
Reviewer2:
I wonder if any DMTs may have effects on astrocyte-derived exosomes.
Answer:
In our study all enrolled participants were treatment-naïve. This was one of the inclusion criteria.
Reviewer2:
How astrocyte-derived exosomes come into the circulation? I wonder if they have any effects on immune cells (T cells) in the CNS.
Answer:
As was mentioned in the discussion section, there are reports pointed that exosomes can migrate across the BBB (doi:10.1016/j.jconrel.2017.07.001). However, the exact mechanism is poorly understood. The primary proposed mechanism is transcytosis. Some studies have suggested that inflammatory environment is needed for EVs migration (topic reviewed in: doi.org/10.1186/s12987-022-00359-3). However, more studies are needed to unravel further details of this mechanism, like in vivo experiments with intravital techniques. We assumed that exosomes can affect T cells’ function not only in the periphery (as was studied by the Dickens et al., 2017, doi:10.1126/scisignal.aai7696), but also within the CNS. However, this issue needs additional experiments in vivo with adequate animal model.
Reviewer 2:
- In Figure 4, no need to mention p=0.07.
Answer:
We have removed all marks
Reviewer2:
Components of astrocyte-derived exosomes should be investigated in details. What about their targets in T cells?
Answer:
We agree with the reviewer. Knowledge about the exosome cargo and its impact on recipient cells is of huge importance. We are planning such analysis in the future.
Round 2
Reviewer 2 Report
The authors have answered my questions.